# EchoPT: A Pretrained Transformer Architecture That Predicts 2D In-Air Sonar Images for Mobile Robotics

**DOI:** 10.3390/biomimetics9110695

**Published:** 2024-11-13

**Authors:** Jan Steckel, Wouter Jansen, Nico Huebel

**Affiliations:** 1Cosys-Lab, Faculty of Applied Engineering, University of Antwerp, 2020 Antwerpen, Belgium; wouter.jansen@uantwerpen.be (W.J.); nico.huebel@uantwerpen.be (N.H.); 2Flanders Make Strategic Research Centre, 3920 Lommel, Belgium

**Keywords:** predictive processing, pretrained transformers, predictive brain hypothesis, sonar data processing, robotic navigation

## Abstract

The predictive brain hypothesis suggests that perception can be interpreted as the process of minimizing the error between predicted perception tokens generated via an internal world model and actual sensory input tokens. When implementing working examples of this hypothesis in the context of in-air sonar, significant difficulties arise due to the sparse nature of the reflection model that governs ultrasonic sensing. Despite these challenges, creating consistent world models using sonar data is crucial for implementing predictive processing of ultrasound data in robotics. In an effort to enable robust robot behavior using ultrasound as the sole exteroceptive sensor modality, this paper introduces EchoPT (Echo-Predicting Pretrained Transformer), a pretrained transformer architecture designed to predict 2D sonar images from previous sensory data and robot ego-motion information. We detail the transformer architecture that drives EchoPT and compare the performance of our model to several state-of-the-art techniques. In addition to presenting and evaluating our EchoPT model, we demonstrate the effectiveness of this predictive perception approach in two robotic tasks.

## 1. Introduction

Robots that navigate the real world often encounter noisy or incomplete sensor data, especially when using low-cost sensing modalities using long wavelengths, such as radar and sonar, and more so in industrial or agricultural environments. Indeed, industrial environments are littered with ultrasonic noise sources [1], such as compressed air [2] and CNC (computer numerical control) machining tools such as lathes and mills [3], and agricultural environments have high noise loads when handling grains, for example, [4], just to name a few. Despite the prevalence of these noise sources, one of the significant benefits of using ultrasound for robotics applications is the fact that they are virtually unhindered by medium distortions such as fog or dust particles [5], as well as the low cost of relatively high-resolution 3D imaging sonar sensors [6,7,8,9]. These sensors can generate 2D or 3D range/direction representations of environments, often referred to as energyscapes [10], and have proven their application potential in mobile robotics. One interesting approach to mobile robotics using ultrasound is based on the subsumption architecture [11] and uses an analog to the widely known optical flow which is called acoustic flow [12,13,14]. Acoustic flow is an approach to using expected transformations in the sensor observations, based on the robot’s ego-motion data, to control a robot in a desired manner without explicit object segmentation.

When analyzing the acoustic flow equations given in [12,13,14], one can interpret these as a special, closed-form case of predictive processing. Predictive processing, or the predictive brain hypothesis, states that perception can be interpreted as the minimization of an error signal between a model-generated perception token and an observed token by altering the model’s state [15,16]. As the acoustic flow model defines how sensor data changes and proposes closed-form solutions for testing predictions of sensor data evolution over time, given a set of robot ego-motion parameters, acoustic flow can be seen as a form of predictive processing. While the acoustic flow model has proven to be a capable paradigm for robot navigation, issues remain with using acoustic flow as a prediction model. As a primary issue, the acoustic flow model ignores the concept of a point-spread function in the imaging sonar, ignoring aspects like the Rayleigh limit, which do govern real-world sonar sensors [17,18,19]. Furthermore, the acoustic flow model introduces deformations of the actual point-spread function when used for sensor data prediction, based on the non-isotropic forward prediction function [13].

To overcome these issues, we propose a learning-based model. This model is based on deep neural networks and transformers to perform sensor data prediction, and it is trained in a completely self-supervised manner, inspired by how large language models are trained [20,21]. This self-supervision allows for the gathering of vast amounts of sensor data without an expensive labeling step or the need for teacher sensor data. Indeed, the introduction of teacher sensor modalities has been performed in previous works [22,23,24,25], in which either image enhancement was trained using a structured environment or a supervisory sensor modality was used (such as a camera or a 3D light detection and ranging, LiDAR) to produce a depth map to which the neural network regresses using sensor data. In our approach, called EchoPT (Echo-Predicting Pretrained Transformer), we use only the information from the sonar sensor, in conjunction with the desired robot ego-motion parameters set via the robot’s controller, to predict future sensor frames, either in a single shot or in an auto-regressive manner (where new predictions are made using previous predictions).

In the remainder of this paper, we will first detail the general architecture of the problem that EchoPT solves and give implementation details on our transformer-based neural network architecture. Next, we will provide exhaustive performance metrics on EchoPT and benchmark its prediction capabilities with several state-of-the-art methods. Then, we will illustrate the power of a system like EchoPT for predictive processing in the context of mobile robotics applications and the advantages that EchoPT offers over existing methods. Finally, we will discuss the limitations of our work, as well as the benefits that the EchoPT model provides, and indicate areas of future work.

## 2. EchoPT—The Echo-Predicting Pretrained Transformer

### 2.1. Problem Formulation and Experimental Setup

As a general concept, the EchoPT model aims to predict a novel sonar image token based on a sequence of previous sonar tokens and associated previous velocity commands executed via the robot and the velocity commands that the robot will perform in the next step. Figure 1 shows an overview of this. The robot and a schematic view of the sonar sensor are depicted in panel (c), showing the linear ($v_{l}$) and rotational ($\omega_{r}$) velocity components of the robot. Three reflectors are in the sensor’s field of view, labeled P1, P2, and P3. Each of these is observed in a polar coordinate system consisting of range *r* and azimuth direction θ. Note that, in this paper, we restrict ourselves to 2D sonar images in the horizontal plane, which has been shown to be sufficient for many basic robotics tasks [26]. Extensions to 3D sonar sensors are currently not considered in this paper, as they require a non-uniform spatial sampling strategy based on the fact that sonar data lives on a non-Euclidean manifold [27]. We will detail this limitation in Section 4.

When the robot performs a particular set of velocity commands, the evolution of the sensor data over time can be calculated using the acoustic flow equations. More specifically, when performing a linear motion (without rotation components), the reflectors move over so-called flow lines, which are the solution to a set of differential equations (see [12]):(1)$$r\left[t\right]\;\cdot \;sin\left(\theta \left[t\right]\right)=R_{c}$$
Here, r[t] stands for the range at which the reflector is observed, and θ[t] is the azimuth direction. The constant $R_{c}$ is the range the flow-line intercepts with the θ=0 direction. Exemplary flow lines can be seen in Figure 1b.

To generate data for the experiments in this paper, we used the simulator developed in [12,13,14], which has been validated to produce life-like sonar data for human-created environments where robot algorithms have been trained in simulation and deployed using real-world sonar sensors without modifications to the processing pipeline. Using a simulation engine allows for the generation of vast amounts of data and repeatable experiments on the effects of the implemented control algorithms, allowing a statistical analysis of the results. The simulation environment in which the robot navigates is shown in Figure 1a. For benchmarking purposes, we implemented three approaches to predicting novel sensor tokens, all shown in panels (d–f). In panel (d), we illustrate the most naive approach: for a given rotational and linear velocity, the images are shifted over the performed displacement along the according coordinate axes. This model is correct for rotational velocities but incorrect for linear velocity components, as seen from the acoustic flow equations in [12]. Furthermore, transmitter and receiver directivities are not taken into account in this approach. A more refined approach is illustrated in panel (e), where we use the acoustic flow equations to predict the novel sensor views by transforming each image coordinate pixel to their new locations and then resampling the image on the original grid. This approach, however, still fails to incorporate spatial directivity patterns. Finally, panel (f) shows the approach using EchoPT, which takes in a stack of *n* previous sonar tokens and a set of previous velocity commands and predicts the next sonar token from that data and the planned velocity commands. Note that the naive and acoustic flow methods only process a single input image, as there is no precise formulation on using multiple input images to perform that prediction with multiple input frames. Panel (g) shows the sensor modeled in the simulation engine, an eRTIS (embedded real-time imaging sonar) sensor [6] that generates the desired range/direction energy maps of the environment.

### 2.2. The Architecture of EchoPT

When designing the architecture of our EchoPT model, we were inspired by two mainstream model families. On the one hand, we took inspiration from large language models using transformers, which perform an auto-regressive prediction of tokens (in our case, sonar images instead of text), trained in a self-supervised manner [28]. On the other hand, we were inspired by vision transformers [29,30,31], which use patch embedding to embed the large input images into a lower-dimension latent space, which the transformer model subsequently processes. Similar approaches using the idea behind ViT have been used on 2D underwater sonar data for object recognition and detection [31,32,33] but not for the prediction of the following sensor token. The overall architecture of our model can be seen in Figure 2. The model consists of three main branches: a branch with a transformer, using patch embedding and positional encoding to embed the input images into a latent space, which then gets concatenated with the velocity commands. This is passed through several transformer layers, each consisting of self-attention (6 heads and 384 key, query, and value channels), and a non-linear transformation with a 500-dimensional latent space. After the transformer step, the patch embedding is reversed using the full transformer latent space, and the resulting 2D images are then passed through several feed-forward convolutional layers. In parallel, a feed-forward 2D convolutional pipeline operates on the stack of input images, and an MLP pipeline operates on the velocity commands. The outputs of these three branches are then resized to a uniform size and depth-concatenated. Finally, these concatenated representations are passed through a convolutional pipeline, yielding the final output image. The model, as implemented, has around 9 million learnable parameters, with the majority (7.8 million) situated in the eight transformer layers. The detailed architecture can be found in the source code posted on Zenodo [34] and in the Appendix A in Table A1. The model is implemented using the Matlab Deep Learning Toolbox, version 2024a, using custom layers for the transformer implementation.

### 2.3. Data Set and Training

As stated before, we chose to build the data set for this paper using the simulator that was developed in [12,13,14]. This simulator was validated with real-world measurements, as it has been used to develop and tune complex robotic control algorithms, which were then transferred to real-world robot experiments without changing the processing pipeline. Furthermore, using a simulator allows the generation of large data sets, as the entirety of data generation can happen in parallel and faster than in real time. For this experiment, we generated ten data sets of 25 min. Each data set contained 7500 sonar measurements measured at 5 Hz, with linear robot speeds between −0.3 m/s and 0.3 m/s and rotational speeds between −1 rads^−1^ and 1 rads^−1^. In total, we simulated 5 h of sonar data for the initial training and test data set, consisting of 75,000 sonar images. Each run consisted of a randomized world (with an example shown in Figure 1a, in which 3750 sonar measurements were taken. For each of the 20 runs, we randomized the world layout and robot starting pose. We kept the acoustic noise levels constant throughout all training runs. Before training, the data set was pre-processed to build the input data stacks, together with the velocity commands, and we used custom file data stores for easy access to the training data.

The model was trained using around 3.5 h of sonar data (around 63,000 sonar images) on a single NVIDIA RTX 4090 GPU (graphics processing unit), and it took around 36 h. We used the Adam optimizer with a constant learning rate of 5 × 10^−5^ for 1000 epochs with a mini-batch size of 64. The final network was picked by choosing the best validation loss after 1000 epochs, calculated on a validation set of 1600 image stacks (three input images, eight velocity commands, and one output image) that were chosen randomly from the original data set. Finally, all tests were performed on an additional simulation run, which was not used during training. Details on the optimizer settings can be found in Appendix A in Table A2. As a loss function, we used the Huber loss between the predicted and the training images. The inference time of one run of EchoPT is around 100 ms on a single RTX4090 GPU. The model was implemented in Matlab with 32-bit floating-point representation for weights and internal states. The model consists of a a custom transformer implementation that is written using the automatic differentiation tools available in Matlab. The custom transformer implementation is then combined with the convolution operations using the Deep Network Designer. The inference time of 100 ms can be significantly increased with more attention to the practical implementation of the model (i.e., fully written in PyTorch, a reduction in internal floating point precision, etc.). However, these optimizations are out of scope in this paper. Indeed, the optimization of neural network architectures for computational efficiency is an extended research field on its own, and extensively optimizing the network parameters would draw attention away from the main message of the paper.

## 3. Experimental Results

In this section, we will detail several experimental results obtained using our EchoPT model. We will also detail several prediction experiments and benchmark them against the naive and acoustic flow method for next-token prediction. Furthermore, we will detail the power of predictive processing using EchoPT in two robotics experiments.

### 3.1. Prediction Performance of EchoPT

To evaluate the capabilities of EchoPT, we performed an experiment where the task of the model was to predict the next sonar sensor token in a sequence of images. Figure A1 in Appendix B shows an example of such a prediction. Figure 3 shows a summary of this. In the figure in Appendix B, the input sequence of images is shown in panels (a–c), indicated with T1, T2, and T3. The following image in the sequence is T4, which is the image that is to be predicted based on T1–T3. The prediction of EchoPT can be seen in panel (e), labeled T4 (Predicted). As the differences between subsequent frames are not that large, we calculated the differences between T4 (predicted) and the images T1–T4 and plotted them in panels (f–i). These difference images show that the difference between T4 and the predicted image is very low (panel (i)) and that there indeed is a difference between T4 (predicted) and T1–T3 due to the robot motion. We calculated the 2D correlation between T4 (predicted) and T1–T4, shown in panels (j–n), to further illustrate the prediction capabilities. Here, the blue cross indicates the middle of the correlogram, corresponding to no shift in the range or direction axes. The difference between the peak of the correlation function and the shift is largest when comparing the oldest images to the prediction and zero when comparing T4 to the prediction (panel (m)). This further illustrates the performance of the applied method.

To benchmark the performance of EchoPT, we compared the predictive performance of EchoPT to two other methods, the naive method and the acoustic flow method, as explained in the previous section. Figure 4 shows an example of this comparison. We predicted the next sonar token using the three previously mentioned methods and plotted the resulting image. We highlighted two features (labeled 1 and 2) in the image, which show the main differences between the two approaches. The naive approach fails to incorporate intensity changes and local deformations (which is especially apparent in feature 1). In contrast, the acoustic flow approach deforms the imaging sensor’s point-spread function (especially visible in feature 2). The point-spread function of an imaging sensor is an essential aspect of the performance of the sensor [35], and altering this point-spread function can have detrimental effects on downstream tasks such as place recognition [36]. In contrast, our EchoPT model successfully predicts the sensor data transformation while retaining all of the major image features.

While the results in Figure 4 are promising, they are illustrative and not evidence of a well-functioning prediction model. To further analyze the performance, we performed token prediction using the three models on 1000 images from the test data set and calculated two performance metrics: the cross-correlation and the normalized root mean squared difference. We then calculated each test run’s mean and standard deviation and collated these data in Table 1. In this table, the first column indicates the one-shot prediction error, which has been tested in this experiment, and we can conclude that the EchoPT model outperforms the other models in a statistically significant manner (two-sample *t*-test, *p* < 0.001).

### 3.2. Auto-Regressive Prediction

We performed an auto-regressive prediction task to test the model’s capabilities further. In this task, the model was tasked with predicting images several time steps ahead, using its previous predictions as input data. We tested three scenarios, AR-3, AR-5, and AR-10 on the same 1000 samples from the test set. The number behind the AR designator indicates how many frames the model needs to predict in the future (i.e., 3, 5, and 10). The one-shot approach would, in that line of reasoning, be AR-1. Exemplary results can be found in Appendix B in Figure A2 and Figure A3, showing the predictions and the prediction differences, respectively. However, Table 1 also shows the performance metrics of the three models for the three auto-regressive prediction tasks. Again, the EchoPT model outperforms the other models in all tasks in a statistically significant manner (two-sample *t*-test, *p* < 0.001).

### 3.3. Predictive Processing: Slip Detection

As having a model that allows for the prediction of sensor data tokens can be a powerful tool (as demonstrated expertly by large language models [28]), we want to demonstrate the possible applications that this approach brings to robotics. In the first application, we used our EchoPT model to detect wheel slips in a mobile robot. Indeed, a wheel slip is a challenging problem for outdoor robotics, where terrain conditions are not always predictable [37,38]. To illustrate the usability of EchoPT in this application, we predicted future sensor frames in an auto-regressive manner using sensor data, velocity commands, and previous predictions. We then calculated an error signal between the predicted frame and the measured frame and calculated an error signal, ϵ(t), as follows:(2)$$\epsilon \left(t\right)=\frac{\sqrt{\sum_{r,\theta }{\left[I_{p}\left(r,\theta \right)-I_{m}\left(r,\theta \right)\right]}^{2}}}{\sqrt{\left|CC(I_{p},I_{m})\right|}}$$
where $I_{p}$ is the predicted image, $I_{m}$ is the measured image, and CC stands for the 2D correlation coefficient. The resulting error signals are calculated for three conditions, one-shot, AR-3, and AR-5, for the three prediction models (naive, acoustic flow, and EchoPT). The results are shown in Figure 5, where each panel shows the results from the different predictors. We introduced two slip conditions, starting at 10 s and 30 s; in the first, both wheels of the robot slipped, and in the latter, only one wheel slipped. While all three detectors detect the first condition well, it is the second condition where EchoPT has the advantage, especially when using the five-time step auto-regressive prediction (which, keep in mind, has no measured data in the final prediction step anymore and relies purely on previously predicted frames). The authors note that other techniques, such as using an IMU (inertial measurement unit), might offer a better solution to this particular problem, but the illustrative purpose of this application still stands. Around the 36 s mark, a discernable spike is present in the naive predictor and acoustic flow predictor, which is absent from most of the predictions made using EchoPT. The spikes in the naive and acoustic flow predictor are caused by specific echo scene conditions where reflectors are located in such a way that prediction using these two methods fails (i.e., driving parallel to a wall without other reflectors being present). As EchoPT encountered these scenarios during training, they are recognized as a valid evolution mode of echoic information. In turn, this causes no faulty predictions and, therefore, no (or a small) spurious spike in the detection signal.

### 3.4. Predictive Processing: Robot Control

A second, more challenging application is using our EchoPT model to overcome failures of sensor measurements due to high noise conditions. In industrial contexts, large bursts of ultrasonic noise might occur, completely overwhelming the faint echoes that the sensor picks up. Other failure modes might include the intermittent loss of communication with the sensor, partial occlusions, etc. We set up an experiment where a robot drove through a corridor of circle reflectors, as shown in Figure 6. These reflectors, indicated as red circles, could represent trees in an orchard or the pillars of racks in an industrial hall. The robot spawns with a random position and orientation in the green rectangular spawn boxes and drives toward the waypoint indicated by the green circle. The trajectory is repeated 50 times, and the robot locations are recorded. The robot uses the control algorithm detailed in [12,13,14], which is based on the subsumption architecture and makes use of the acoustic flow model to perform corridor following. In panel (a), we show the kernel density estimates of the robot trajectories for all 50 runs. The robot follows a stable trajectory through the corridor, further illustrated by the distribution of travel times (panel (d)) and the distribution of deviation from the middle of the corridor (panel (e)).

Next, we introduce a sequence of high noise bursts, rendering the sensor data unintelligible during periods of 1.2 s (see panel (f)). The curve in panel f shows that, approximately 30% of the time the robot drives, the data are lost due to the noise injection (changing the SNR from 5 dB to −80 dB). The resulting robot trajectory is shown in panel b, which shows the robot stopping due to the absence of sensible sensor data (indicated by the nodules in the kernel density estimates, as these correspond to the robot standing still). This is further illustrated by the significant increase in travel time (panel (d)) and the much larger deviation from the midline (panel (e)), indicating that the robot controller encounters significant difficulties following a stable path through the corridor.

To overcome this problem, we applied the EchoPT model in auto-regressive mode to predict sensor data during the noise bursts, based on previous measurements and the model’s predictions. During the noise bursts, we fed the predicted sensor tokens from EchoPT into the robot controller, unaware of the noisy sensor conditions. The resulting robot paths are much more stable (panel (c)), approaching the performance of the noiseless case. This is also shown in panel (d), with only a slight increase in transit time, and in panel (e), with only a slight increase in deviation from the middle of the corridor.

## 4. Limitations

When contemplating the implementation of EchoPT and the experimental data illustrating its performance, it is not hard to come up with some limitations of our current presentation. For one, we do not use the concept of transformers in our system to its full potential. As the EchoPT model is working with a sequence of images, it would be wise to look for inspiration on how transformer architectures for video are constructed [39,40,41] by utilizing a two-tier transformer architecture, one for temporal information and one for spatial information. This is something we will be addressing in our future work, as this is a natural extension of the auto-regressive mode we have now implemented.

Another shortcoming is the use of 2D convolutions in the processing stack. When extending this idea of sensor data prediction to 3D, one inevitably stumbles into the fact that sonar data occupy a spherical manifold instead of a Cartesian one. Assuming a third dimension of the elevation direction and using a Cartesian grid to represent the data leads to severe over-representation of the data around the sphere’s poles. Therefore, 2D convolutions are not a suitable operator to represent transformations in the 3D case. Solutions could involve replacing the 2D convolutions with operators using spherical harmonics as basis functions [42,43] or replacing the convolutional blocks altogether with a transformer and using spatial patch embeddings that make use of patches distributed uniformly over the unit sphere with an equal area [44,45,46].

In this study, we did not present any ablation experiments on the network architecture, which can be attributed to the limited availability of computation resources, on the one hand, as well as the central message of this paper. Indeed, the main message is that transformer-like architectures can be used to perform predictive processing on in-air sonar data (which, to the best of the authors’ knowledge, is the first time this has been achieved). As we do not claim that the architecture of the EchoPT model is ideal in any way, the execution of ablation experiments was not part of the experiments in this paper. In terms of computing, a single training run currently takes around 36 h, and a statistically meaningful ablation study would take around several weeks to compute, which is not available to the researchers at the time of writing. The sizes of the internals of the transformer architecture (i.e., number of key, value, and query triplets, and the size of the MLP pipelines) were chosen based on “good practices” gained from other literature and seeing other people’s work. For the convolutional steps, the convolution sizes were chosen to have an acceptable receptive field over the input images (i.e., not too small, so that only a small number of range/direction cells is included in the convolution operators, and not too large, so that the resulting convolutions would, in fact, be a fully connected linear operation). These judicious choices resulted in the approximately 9 million parameters of the final network architecture.

A final shortcoming of the current paper is the lack of real-world experiments, as all calculations were performed using a simulator. This simulator was already used extensively in other papers [12,13,14], including the validation of this simulator with real-world experiments. Therefore, we believe using a simulator in this study is, instead, an advantage because it allows us to produce much more data and many more experiments, which then allows us to perform a detailed statistical analysis of the experimental data. Real-world experiments are also part of our future work.

## 5. Conclusions

In this paper, we have presented our new EchoPT model, which stands for the Echo-Predicting Pretrained Transformer. We introduced the concept of predictive processing in robotics and the merits that this approach can bring to robotic perception in complex sensing scenarios. We thoroughly evaluated the performance of our EchoPT model in sensor token prediction tasks, both in single-shot and auto-regressive mode, and compared it to two other methods that solve the same prediction problem. The extensive tests we performed to quantify the performance of the models were analyzed using the appropriate statistical methods, and they indicate that our EchoPT model outperforms the benchmark models in a statistically significant manner.

In addition to solving sensor token predictions, we applied predictive processing in two robotic use cases: robot wheel-slip detection and robotic motion control in high-noise environments. In the wheel-slip detection, we built a detector signal that allows for the detection of inconsistencies in sensory flow when the robot is not adhering to the underlying motion model (indicating a wheel slip). Our EchoPT model outperforms the other two benchmark models in this task, and it is the only model that robustly detects the single-wheel slip condition. In the second task, a robot was challenged to navigate a corridor environment, where 30% of the time, the robot’s perception was rendered uninformative through the addition of vast quantities of noise (from 5 dB SNR to −80 dB SNR). Using auto-regressive prediction models based on our EchoPT prediction, the robot can still robustly navigate the corridor environment in which the robot is placed. The performance of this system was tested using 50 runs in each case.

We believe that, with this paper, we have indicated the advantages deep neural network architectures based on transformers can offer in supporting predictive processing for neglected sensor modalities such as in-air sonar sensing. Indeed, the number of researchers working on vision or LiDAR-based systems for robotics is vastly greater than that of researchers working on in-air ultrasound for robotics (at the time of writing, searching for “robotics sonar ultrasound air” on Google Scholar since 2023 yields 425 papers, while searching for “robotics vision LiDAR” yields 16,800 results). We hope that, by writing this paper, we can inform the machine learning community about this powerful modality, which is being used daily by millions of bats worldwide.

## Figures and Tables

**Figure 1 biomimetics-09-00695-f001:**
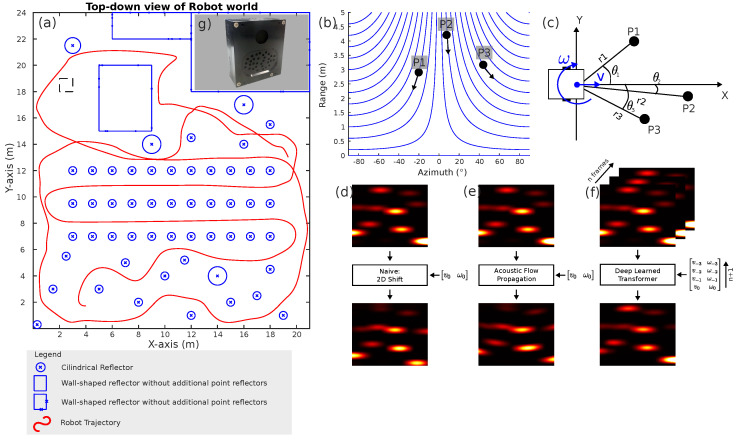
Overview of the experimental setup. Panel (**a**) shows the simulation environment in which a two-wheeled robot drives. A sketch of the robot is shown in panel (**c**). The robot uses an array-based imaging sonar sensor panel (**g**) capable of generating range-direction energy maps (called energyscapes), shown in panels (**d**–**f**). This sensor is modeled in the simulation environment based on accurate models of acoustic propagation and reflection. Panel (**b**) shows what is called the acoustic flow model. This model predicts how objects in the sensor scene move through the perceptive field based on a certain robot motion. The blue flow lines are shown for a linear robot motion. Panels (**d**–**f**) show the task that is being solved in this paper: how can novel sensor views be synthesized given a certain set of robot velocity commands $\left[\begin{array}{cc} v_{lin} & \omega_{r} \end{array}\right]$? Each of these velocities has a time-step index, as shown in panels (**d**–**f**). Panel (**d**) shows the prediction based on the naive shifting of the image in the range and direction dimensions. Panel (**e**) shows the operation using the acoustic flow model of panel (**b**). Both of these operators can only use the last frame to perform the prediction. Panel (**f**) shows the EchoPT model, which takes in *n* previous frames and velocity commands and predicts the novel view using a transformer neural network.

**Figure 2 biomimetics-09-00695-f002:**
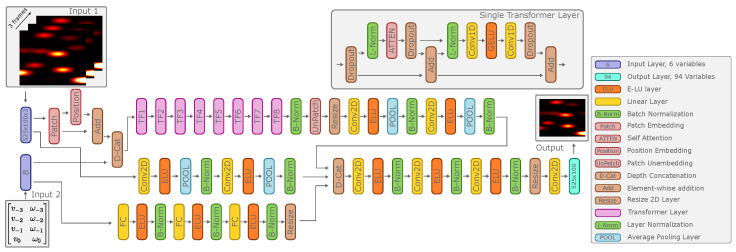
Overview of the network architecture of EchoPT. The EchoPT model has two inputs: the set of *n* previous input frames (set to three in this paper) and the n+1 velocity commands (three previous and one for the prediction). The model has three main parallel branches: a transformer branch, a feed-forward convolutional branch for the sonar images, and an MLP (multi-layer perceptron) pipeline using the velocity commands as input. These three branches are depth-concatenated and passed through more feed-forward convolutional layers to obtain a single output image.

**Figure 3 biomimetics-09-00695-f003:**
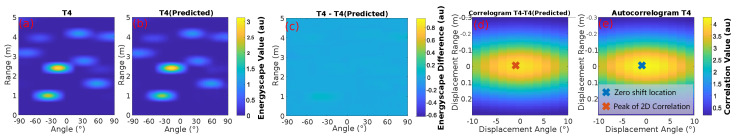
Condensed version of Figure A1 in Appendix A. Panel (**a**) shows the target sonar image, and panel (**b**) shows the predicted image. Panel (**c**) shows the difference between the two images, and panels (**d**,**e**) show the 2D correlogram.

**Figure 4 biomimetics-09-00695-f004:**
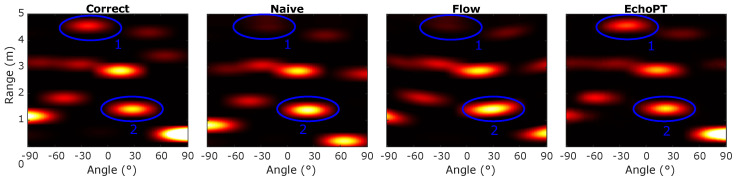
Prediction results of a single frame using three prediction methods: the naive operation, which shifts the image in the range and direction dimensions; the acoustic flow approach, which uses the acoustic flow equations to transform the image; and finally, the EchoPT prediction.

**Figure 5 biomimetics-09-00695-f005:**
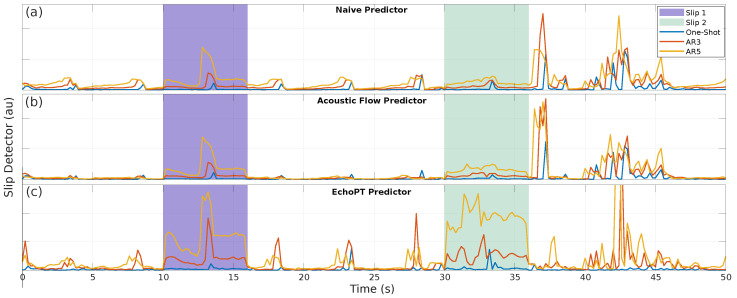
A first application of predictive processing in which a robot performs a trajectory in the environment from Figure 1. In two periods (between 10 s and 16 s and between 30 s and 36 s), the robot encounters slip conditions (meaning the robot is not performing the motion that the robot expects to perform). In the first section, the robot is slipping on both wheels; in the second condition, only one wheel slips. The plots show the slip detector, which uses differences in the predicted and measured sensor data for different prediction horizons (one-shot, three-frame auto-regressive, and five-frame auto-regressive). Longer time horizons provide the clearest slip detection signal, with EchoPT being the only one that detects the second slip condition. Panel (**a**) shows the results for using the naive predictor, panel (**b**) for the acoustic flow predictor and panel (**c**) for the EchoPT predictor.

**Figure 6 biomimetics-09-00695-f006:**
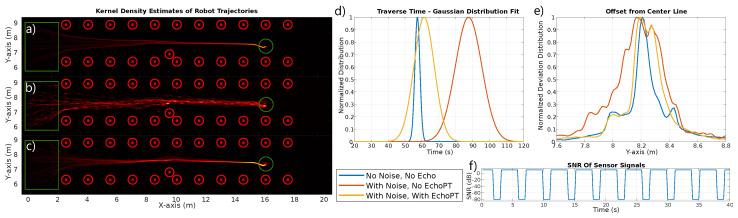
A second application of predictive processing in which a robot is tasked with driving from the green rectangular spawn boxes to the waypoint indicated by the green circles, using a subsumption-based control stack described in [13]. Panel (**a**) shows the kernel density estimate of 50 runs with clean sensor data (signal-to-noise ratio, SNR = 5 dB). In panels (**b**,**c**), we added intermittent noise to the measured sensor data (shown in panel f, SNR = −80 dB). In panel (**b**), the original controller was used, showing the traversed paths’ deterioration. In panel (**c**), sensor data were predicted in an auto-regressive manner using EchoPT for the duration of the noise bursts and fed into the controller instead of the noisy data. Panel (**d**) shows the travel time for the robot in the three conditions, showing a large increase in travel time for the controller from panel (**b**). Panel (**e**) shows the deviation from the midline of the corridor, again showing a large deviation when no predictive processing is used. Panel (**f**) shows a small section of the evolution of the SNR over time.

**Table 1 biomimetics-09-00695-t001:** Performance metrics for sonar prediction.

	Cross-Correlation [ μ(σ( ↑ ]			
**Method**	**One-Shot**	**AR-3**	**AR-5**	**AR-10**
Naive	0.85 (0.24)	0.66 (0.31)	0.56 (0.30)	0.38 (0.29)
Acoustic Flow	0.86 (0.24)	0.73 (0.33)	0.69 (0.34)	0.56 (0.33)
**EchoPT (ours)**	**0.97 (0.02)**	**0.83 (0.25)**	**0.80 (0.26)**	**0.69 (0.24)**
	**Normalized Root Mean Squared Difference [** μ(σ) **↓]**			
**Method**	**One-Shot**	**AR-3**	**AR-5**	**AR-10**
Naive	0.42 (0.37)	0.76 (0.54)	0.83 (0.44)	1.34 (3.22)
Acoustic Flow	0.38 (0.37)	0.62 (0.56)	0.65 (0.61)	0.99 (1.63)
**EchoPT (ours)**	**0.15 (0.11)**	**0.46 (0.45)**	**0.52 (0.31)**	**0.80 (1.07)**

## Data Availability

We provide the source code and a data set used in this study on Zenodo under [34].

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
