# Peer review of "EchoPT: A Pretrained Transformer Architecture That Predicts 2D In-Air Sonar Images for Mobile Robotics"

_biomimetics, 2024, doi:10.3390/biomimetics9110695_

Round 1

Reviewer 1 Report

Comments and Suggestions for Authors

1) In the " Abstract.", for the first occurrence of these abbreviations "EchoPT", it needs to list the full name.

2) In the " 1. Introduction", for the first occurrence of these abbreviations "CNC",”LiDAR”, it needs to list the full name. And in line 110,”eRTIS”? And in line 130,MPL? And in line 152,”NVIDIA RTX 4090 GPU”?And in line 255,”SNR”?

3) The font in the “Figure 3.”is too small to read easily.

4) In line 158, the table serial numbe is not marked correctly.

5) In Fig. 1,r1、r2、r3、  θ1θ2θ3、υ0、υ_1、υ_2、υ_3、ω0、ω_1、ω_2、ω_3 are not mentioned in this paper.

6) In the section 3.3,for the first occurrence of these abbreviations "SNR",it needs to list the full name.

7) On line 234, IMU needs to list the full name.

Comments on the Quality of English Language

Moderate editing of English language required.

Author Response

1) In the " Abstract.", for the first occurrence of these abbreviations "EchoPT", it needs to list the full name.

Done

2) In the " 1. Introduction", for the first occurrence of these abbreviations "CNC",”LiDAR”, it needs to list the full name. And in line 110,”eRTIS”? And in line 130,”MPL”? And in line 152,”NVIDIA RTX 4090 GPU”?And in line 255,”SNR”?

Done

3) The font in the “Figure 3.”is too small to read easily.

Increased Font size

4) In line 158, the table serial numbe is not marked correctly

This has been corrected

5) In Fig. 1,r1、r2、r3、  θ1、θ2、θ3、υ0、υ_1、υ_2、υ_3、ω0、ω_1、ω_2、ω_3 are not mentioned in this paper.

We added an explanation of these symbols in the caption

6) In the section 3.3,for the first occurrence of these abbreviations "SNR",it needs to list the full name.

Done

7) On line 234, IMU needs to list the full name.

Done

Reviewer 2 Report

Comments and Suggestions for Authors

The authors have introduced a novel architecture, termed EchoPT, which leverages a pretrained transformer model to predict sonar images. This model diverges from traditional algorithms that predominantly depend on the immediate preceding frame, by incorporating multiple previous tokens to enhance prediction accuracy. Simulation results are provided to demonstrate the superior predictive performance of EchoPT under various conditions. However, prior to publication, several aspects require further clarification:

1) Given the prolonged duration of the training phase, it would be beneficial for the authors to present a graph of the loss function versus epochs to substantiate the necessity of extensive training. Moreover, the justification for utilizing 9 million parameters for this task should be addressed to ascertain its efficiency.

2)In Figure 5, the authors assert that EchoPT was the only model successful in detecting the second scenario where only one wheel slipped. However, it looks like near 36s, there is a discernible spike in the two comparison algorithms which is absent in the EchoPT prediction. The authors are suggested to investigate and explain why this happens and if it can be used as evidence of the slip conditions for the other two algorithms.

Author Response

1) Given the prolonged duration of the training phase, it would be beneficial for the authors to present a graph of the loss function versus epochs to substantiate the necessity of extensive training. Moreover, the justification for utilizing 9 million parameters for this task should be addressed to ascertain its efficiency.

Sadly, we did not save the graph of the loss over training epoch. While we agree that it would be interesting to add it to the paper if it exists, we do believe that rerunning the training process solely to obtain this graph is a waste of electrical energy.

For the amount of learnables: this is a small number compared to state of the art networks in machine learning, which are 4 to 5 orders of magnitude larger. The only justification that one  can give for these numbers is by performing ablation studies. Ablation studies require significant compute, as you need to do a full factorial expansion of several parameters of the model (like transformer sizes). This is out of scope for the current paper, which is already a significant body of work as it stands. For future work we will indeed be doing these kind of ablation studies. We explained the lack of ablation in more detail in the limitations section of the paper.

2)In Figure 5, the authors assert that EchoPT was the only model successful in detecting the second scenario where only one wheel slipped. However, it looks like near 36s, there is a discernible spike in the two comparison algorithms which is absent in the EchoPT prediction. The authors are suggested to investigate and explain why this happens and if it can be used as evidence of the slip conditions for the other two algorithms.

We added explanation for this in the discussion of the result.

Reviewer 3 Report

Comments and Suggestions for Authors

The paper EchoPT: A Pretrained Transformer Architecture that Predicts 2D In-Air Sonar Images for Mobile Robotics introduces a novel transformer-based architecture designed to predict future 2D sonar images using previous sonar data and robot motion commands. The model is trained in a self-supervised manner and evaluated through simulated experiments, demonstrating its effectiveness in prediction tasks and potential applications in mobile robotics, such as slip detection and noise handling.

  • Page 3, Figure 1(a): The simulation environment depicted in (a) is not clear. It would be helpful to include a legend to enhance clarity.
  • Page 5, line 148: “In total, we simulated 5 hours of sonar data for the initial training and test dataset, consisting of 75,000 sonar images.” There is a lack of detailed explanation regarding the diversity of these datasets. Please provide more information about the variety of environments, obstacle configurations, and noise levels considered in the training data.
  • Page 5, line 151: “The model was trained using around 3.5 hours of sonar data (around 63,000 sonar images) on a single NVIDIA RTX 4090 GPU and took around 36 hours.” This long training time raises concerns about the model’s computational efficiency. Consider including a section discussing the model's efficiency in terms of inference time, memory usage, and scalability.
Comments on the Quality of English Language

  • Page 1, line 16: "Robotds that navigate the real world often encounter noisy or incomplete sensor data." The word "Robotds" is a typo and should be corrected.
  • Page 5, line 158: “Details on the optimizer settings can be found in the appendix in table ??” The reference should be corrected to "Table 3" instead of "Table ??."

Author Response

Page 3, Figure 1(a): The simulation environment depicted in (a) is not clear. It would be helpful to include a legend to enhance clarity.

Added a legend to the figure

Page 5, line 148: “In total, we simulated 5 hours of sonar data for the initial training and test dataset, consisting of 75,000 sonar images.” There is a lack of detailed explanation regarding the diversity of these datasets. Please provide more information about the variety of environments, obstacle configurations, and noise levels considered in the training data.

The reviewer is correct about the lack of information! Added explanation about the configuration of obstacles, environmnets, and noise levels.

Page 5, line 151: “The model was trained using around 3.5 hours of sonar data (around 63,000 sonar images) on a single NVIDIA RTX 4090 GPU and took around 36 hours.” This long training time raises concerns about the model’s computational efficiency. Consider including a section discussing the model's efficiency in terms of inference time, memory usage, and scalability.

In practice, that is not an extremely long training time when comparing this to other state of the art networks. We added a small section explaining the inference time (100ms per run), which is not optimized. We added which optimizations can be done, and argued why these optimizations are out of scope for this paper.

Page 1, line 16: "Robotds that navigate the real world often encounter noisy or incomplete sensor data." The word "Robotds" is a typo and should be corrected.

Done

Page 5, line 158: “Details on the optimizer settings can be found in the appendix in table ??” The reference should be corrected to "Table 3" instead of "Table ??."

Done

Round 2

Reviewer 1 Report

Comments and Suggestions for Authors
  1. In the "Table 1. Performance Metrics for Sonar Prediction",What does this symbol µ(σ) represent?  How to interpret the meaning of the value 0.85 (0.24) in the table?
  2. In part "3.2. Auto-regressive Prediction",  “AR-3, AR-5, and AR-10”, represents different testing scenarios. What is the full name of AR?
  3.  InFigure 6, What does the red dot in the middle of the red circle in Figure 6 represent? There is no explanation in the text.
Comments on the Quality of English Language

Moderate editing of English language required.

Author Response

  1. In the "Table 1. Performance Metrics for Sonar Prediction",What does this symbol µ(σ) represent?  How to interpret the meaning of the value 0.85 (0.24) in the table?

This has been adressed. µ = mean, sigma = std deviation, AR = autoregressive. We added a line in the table

  1. In part "3.2. Auto-regressive Prediction",  “AR-3, AR-5, and AR-10”, represents different testing scenarios. What is the full name of AR?

AR = Auto-Regressive. We added this in the text.

  1.  In“Figure 6”, What does the red dot in the middle of the red circle in Figure 6 represent? There is no explanation in the text.

Added explanation. It is a cilindrical reflector object.